# A mix-method investigation on acculturative stress among Pakistani students in China

Cao Shan[1], Mudassir Hussain[2]*, Ghulam Raza Sargani[3]

1 School of Education, Minnan Normal University, Zhangzhou, China, 2 School of Education, Huazhong University of Science and Technology, Wuhan, P.R. China, 3 School of Economics and Management, Huazhong Agriculture University, Wuhan, P.R. China

* mudassir2017@hust.edu.cn

**Data Availability Statement:** All relevant data are within the manuscript and its Supporting Information files.

**Funding:** The authors received no specific funding for this work.

## Abstract

This article investigates acculturation stress among Pakistani students who are studying in Chinese universities, located in five provinces where international students are concentrated, with a mix-method approach. 203 students among 260 questionnaire recipients responded the online survey. When using the ASSIS (Acculturation Stress Scale for International Students) as instrument, the Principal Component Analysis Method and SPSS 20.0, we found that Pakistani students are under acculturative stress, 68.53%, 10.97% and 9.15% of them perceived discrimination, home sickness and perceived hate, and 5.25%, 3.11% and 2.58% of them fear, culture shock and guilt respectively. The qualitative segment of the study is consisted of 20 Pakistani students studying in 4 universities located in Wuhan city of Hubei capital enquiring through semi-structured interviews. The findings illustrate that Pakistani students in China are expressing their major concerns on culture shock, home-sickness, food and language barriers while disconfirm ASSIS findings like perceived discrimination, hate, fear and guilt as factors responsible for acculturative stress. The study suggested that pre-departure orientation lectures about host country's cultural values and campus environment, and on-campus extra-curricular, cultural activities and maximum social interaction with local students can effectively acculturate students in new cultural setting, and can lower their acculturative stress.

## Introduction

International students in Chinese universities belong to different countries with different educational backgrounds. These students compose a complex and culturally diversified society by the provision of varied cultural values, mental outlooks and social norms. It has turned into a globalized source of generating monetary and non-monetary revenues from abroad through the enrollment of international students. According to the Project Atlas 2016–2017 China holds the 3rd position among the countries hosting international students with a number of 442,773 international students, United Kingdom is at the 2nd position with 501,045 whereas United States has the top position hosting 1,078,822 international students.

The major population of the international students in China higher education is composed of Asian students. Besides, a considerable number of international students are enrolled from

**Competing interests:** The authors have declared that no competing interests exist.

Middle East and Arab states, e.g. China hosts a number of international students from African countries and Europe as well. According to the Ministry of Education, China statistics 2016–2017, the top ten of major population of international students in China by country of origin is as given as South Korea, USA, Thailand, Pakistan, India, Russia, Indonesia, Kazakhstan and Japan, and African countries as a group inserting the top 2nd. Chinese Universities are experiencing more complex and a diverse student population and China is playing a leading role in Asia hosting international students. Cross-border and regional collaborations have strengthened higher education in Asia [1] and these Higher Education Institutions (HEIs) are focusing on different management and international students' mobility [2].

According to the statistics released by Ministry of Education China dated 2018.03.01, 442,773 international students from 205 countries and regions study in 829 institutions of higher education and of scientific research in 31 provinces in 2016–2017. This number of international students are growing annually and 11.35% increase in 2016 compared to that in 2015. According to the MOE, a growing number of foreign students are choosing to study in China for a master's or Ph.D. degree across a widening range of disciplines, and scholarships granted by the Chinese government are playing an increasingly important role in attracting international students. In 2017, 58,600 foreign students from 180 countries were awarded Chinese government scholarships, accounting for 11.97% of the total in 2017. 88.02% of the scholarship recipients were degree students (51,600); 69.57% (40,800) were master and doctoral students, marking an increase of 20.06% compared to 2016. The self-funded students were 430,600 which accounts for 88.03% of all overseas students. 48.45% of these students were enrolled for liberal arts degrees, while the number of students majoring in engineering, management, science, art and agriculture increased by 20% annually.

International students according to MOE are located in different provinces and cities of China, the top nine concentrations are Beijing, Shanghai, Jiangsu, Zhejiang, Tianjin, Liaoning, Guangdong, Shandong and Hubei. See (Fig 1). The quantitative segment of the study choses 5 major locations of international students including Beijing, Shanghai, Jiangsu, Zhejiang and Hubei where the research team is located, while the quantitative segment of the study employed semi-structured interviews of 20 Pakistani students studying in four universities Wuhan, Hubei based on purposive sampling technique.

The recent One Belt One Road Initiative boosted up Pakistani students' mobility besides all the countries along One Belt One Road Project. Thus, China attracts more international students by providing them with free and quality education in Chinese universities.

This study is focusing on how Pakistani students adjust themselves to Chinese society with a different language, academic system and culture. Acculturation brings various challenges and students have to bear these challenges without the support of family members and friends,

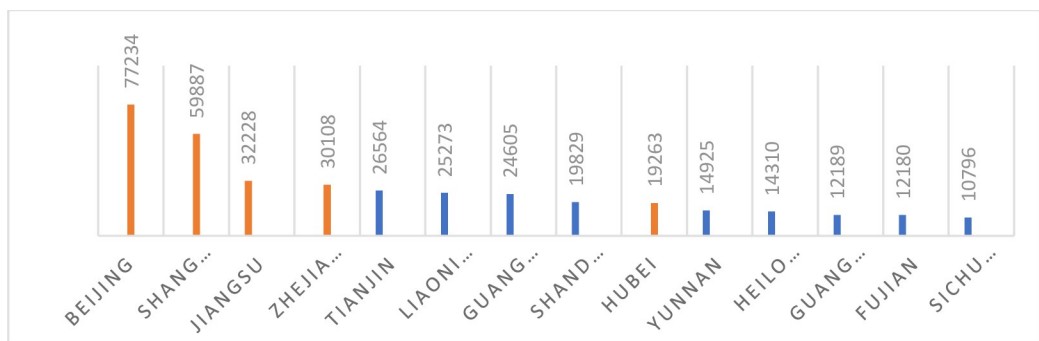

**Fig 1. International students in China by province, 2016–17.**

and they suffer from home sickness. According ASSIS model designed by [3], migrating from their home usually results a stress, missing friends and homes. This mix method study aimed at examining acculturative stress among Pakistani students in Chinese universities using ASSIS scale through online survey and further investigating the real-life experiences of adaptation in China by conducting semi-structured interviews of 20 Pakistani students in 4 universities in Wuhan, Hubei.

## Research questions

The study's aim is to answer the given research questions:

- Are Pakistani students studying in China going through acculturative stress?

- What is the level of perceived acculturative stress of Pakistani students in China?

- What is the role of language in causing acculturative stress?

- How do these students express their experiences while adopting to Chinese society?

## Literature review

Acculturation by term is referred to the extent to which a person from another culture has to learn as language, custom and cultural values like the person of the host culture [4]. International students are expected to adapt the ways of life and basic customs of the people of the host culture such as language, food, dress and cultural values. [5] defines "Acculturation" that it is a process that includes psychological and cultural changes between cultural groups; Several factors impact the process of acculturation, for instance, cultural identity, duration of stay in host country, language competency, social interaction, and engagement with the people of the host country. In acculturation process cultural changes and psychological both ensue within the two distinct cultures but major changes are expected in the minor group and the dominant group usually undergo less changes [6]. Sometimes this process obliges as a mutual adaptation process where two different interacting groups adapt the cultural values of each other. Acculturation has long been focused in other subjects like anthropology and psychology but globalization is a recent phenomenon which provided an opportunity of a close intercultural contact, consequently it led to individual and cultural changes [6].

Research studies illustrate that greater acculturation leads to individual's psychological well-being and health, although the process of change concomitant with acculturation is often perceived as stressful [6]. From a conceptual perspective, acculturation level defines acculturation stress and culture shock, it is significant to make a clear difference between acculturation level and acculturation stress [7]. The international students level of acculturation can be evaluated from how they are acculturated and what is the level of acculturation and how stressful was this process of adjustment while they were passing through, the term is generally put into use for both the phenomena, acculturation process and the considered stress due to acculturation but very rarely the attention is paid to the relation of these two situations [8]. If a student is experiencing difficulties in a new culture and has a low level of acculturation, the student is expected to be probably experiencing more challenges and with a high acculturative stress level. Several aspects of acculturation process of international students' raise adjustment problems that are stressful physically, socially and psychologically [9]. Such a cluster of stress of diverse nature occasionally becomes so intense, and ultimately hinders the adjustment and acculturation process, for they leave the international students with anxiety and disorientation [9]. These students likely to experience less acceptance, tolerance and knowledge of their

cultural practices by the people of the host culture, and sometimes, they may experience even racial discrimination [10].

Low acceptance and tolerance lead an individual to a deviant personality with a negative perceptions and cultural ethnocentrism. Students assume that their own cultural practices are better than that of the host culture. The multiple demands international students experience are the cognitive, emotional and physiological level, when they migrate to another culture [11] are certainly vital for the adjustment of international students and often lead international students to acculturative stress which consequently provokes initial intrapersonal and interpersonal issues. However, these insignificants are neither inevitable for all international students nor are they necessarily the only results of this potentially positive transformation [12]. International students sometimes show discrimination towards the host nationals and other international students of different race, language, color and religion. These factors can add to international students' feelings of loneliness, alienation, powerlessness and depression [9]. International students with a label of being "foreigners" put them into a stressful experience in bending to the host culture and live a life of an exile and their success has a direct relation with the level of their adjustment and acculturation process. These related issues of adjustment seem to lead the literature [13], highlighted depression, loneliness, and homesickness as the dominant concerns of international students. [14] hypothesized the major issues of international students as anxiety, [15] emphasized major concern on stress, fear, pessimism and frustration, but the findings of all these psychological studies on the acculturation process of students are varied and aimless in nature. The newly arrived Pakistani students to China not only need to compete with their Chinese classmates and other international students but they have also to face the challenges of acculturation due to adjustment to the values and practices of the host culture. This group of international students are under a pressure of foreign demands, for instance, the pursuit of academic excellence, learning the cultural values and especially the host community language etc. Somally suggested the term "language shock" to be added to the conceptualization of culture shock for it leads to acculturative stress, and the language holds a dominant position in prompting this phenomenon [16].

[17] identify academic pressure, relationship problems, stereotyping, prejudice, familial concerns, and adjustment to the host country are some of the demands faced by international students. Many researchers have come up with findings like loss of identity [18], and sometimes the acculturative stress develops so intense that it leads an individual to develop a sense of inferiority [3]. [19], for example, while classifying the challenges that are particularly concerned to the international students (language, diet, teaching methodology, finance etc.) and to those of color (racism), the research recognizes issues that are common to all (registration, major selection, and class size in particular). The level of acculturation stress may be varied among international students having different demographic backgrounds and personality types. It is illogical to assume that all the international students experience acculturative stress in the same way and to the same degree [20]. Previously a study, cofounded by Wayne State University USA with the collaboration of Wuhan University China, titled, "Acculturative Stress and Influential Factors Among International Students in China: A structural dynamic perspective" [21], worked on a stratified sample numbered 567 based on the categories of international students in Wuhan city, found that acculturative stress was more common among international students in China than in developed countries and was also more common among international students who did not well prepared, married, and belonged to an organized religion.

[40] conducted a study on "The Intercultural Adjustment of Pakistani Students at Chinese Universities." They recruited 15 Pakistani students studying in Huazhong University of Science and Technology, P. R. China to evaluate the effective strategies that promote the

intercultural adjustment and their perceptions of cultural adaptation. [40] found that Pakistani students are satisfied with their educational experience, life experience and learning achievements. Furthermore, Pakistani students in China showed better satisfaction in their social life without feeling any danger or threat. Similarly [41] aimed to investigate the intercultural adaptation of Pakistani students in Chinese universities regarding psychological adaptation and socio-cultural adaptation, through the data analysis of a survey and the interpretive research on students' personal reports. The study found that 87% of the Pakistani students are satisfied with their interaction with the co-nationals and local students, and they are enjoying sense-making experience in China. The study found that language barriers, relationships with professors, staff and peers, the process of learning, a different time schedule and even different food are additional problems cast onto their intercultural adaptation. As for socio-cultural adaptation, it appears that 89% of the respondents' adaptation is positively correlated with the length of stay in China. This study purely targeting Pakistani students by employing mixed method to deeply investigate acculturative stress among Pakistani students in China.

## Research design

This mixed method research design employing concurrent triangulation strategy to examine acculturative stress among Pakistani students studying in Chinese Universities. To compute statistically the level of stress, "ASSIS" Acculturative Stress Scale for International Students was administered through online survey. To investigate in-depth real-life experiences qualitatively and confirm the quantitative findings, 20 Pakistani students studying in 4 universities located in Wuhan are interviewed by employing semi-structured interview technique.

**Consent & participants of the study.** Ethical approval was obtained from the School of Education Research Ethics Committee (Huazhong University of Science and Technology, Wuhan China). Before data collection all the eligible respondents of both survey and interview participants were informed about the aims of the study, voluntary participation, the right to withdraw at any time without giving a reason, and were assured of the confidentiality of the information to be collected and that the research will purely be used of academic purposes. The statement of consent was included in written on both the ASSIS Scale for survey and on the interview protocols. During qualitative data collection, the researcher shared the participants verbally in their native language (Urdu) the aims and objectives of the study.

For quantitative segment of the study, as convenient sampling method 5 major destinations were focused (Beijing, Shanghai, Jiangsu, Zhejiang and Hubei) which are major host locations for international students, See (Fig 2). The total filled questionnaires number 214 out of 260 (82.30%) were received but among total received questionnaires 11 were excluded due to missing values, thus the total effective samples for the study numbered 203. The participants consisted of male 152 (74.9%) and female 51 (25.1%); undergraduate 25 (12.3%), Masters 57 (28.1%) and PhD 121 (59.6%). The participants were also categorized on the basis of discipline as natural sciences 50 (24.63%), social sciences 58 (28.57%), engineering 59 (29.06%), medical 26 (12.80%). The participants are varied on the basis financial status as CSC scholarship (Chinese Government scholarship) 173 (85.22%), HEC (Pakistan) scholarship 14 (6.89%) and self-finance scheme 16 (7.88%). See Table 1

For qualitative segment of the study, 20 Pakistani students studying in 4 universities in Wuhan, Hubei are recruited as purposive sampling technique for interviews. These 20 students belong to different disciplines, level of education and financial status. see Table 2

**Instruments.** In quantitative segment of the study, ASSIS [3, 22] was used as instrument to investigate acculturative stress among Pakistani students. ASSIS 7 subscales were further extended to their particular items as subscale, Perceived Discrimination (8 items; e.g. I feel

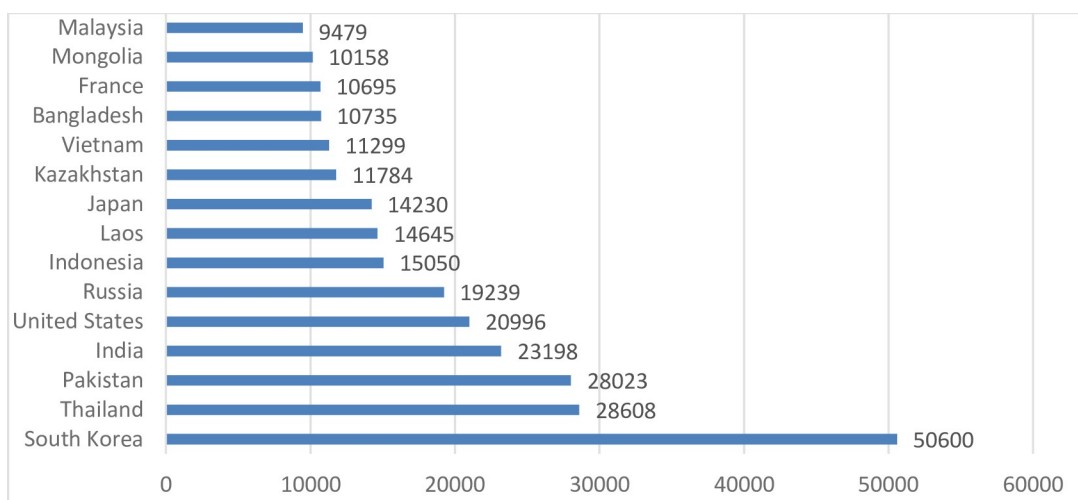

**Fig 2. International students by country of origin in China, 2016–17.**

many opportunities are denied to me), Home Sickness (4 items; e.g. I miss the people and country of my origin), Perceived Hate (5 items; e.g. Others don't like my cultural values), Fear (4 items; e.g. I feel insecure here), Culture shock/ stress due to change (3 items; e.g. I feel uncomfortable to adjust to new food), Guilt (2 items; e.g. I feel guilty to leave my family behind), and Miscellaneous (10 items e.g. I feel nervous to communicate in Chinese language).

In Qualitative segment of the study, semi-structured interviews were conducted in 4 universities in Wuhan, Hubei. Purposive sampling method was applied and 20 students belong to different disciplines, level of education and financial status were recruited. Purposive sampling method is used when the goal of the research is to understand and describe a particular group in depth [23]. The language of communication was used Urdu, the national language of Pakistan for an effective communication. Thematic analysis was made on the explored themes on the semi-structured interview transcript. The recurrent themes were coded under the main themes and relevant themes were coded as sub-themes. The codes were manually coded and color-coding technique was used [24] to understand the highlighted themes. Then the theme had been allotted acronyms for data analysis.

**Testing reliability and validity of ASSIS scale.** Each item was answered on 5 Point Likert Scale, as 1 strongly disagree to 5 strongly agree. The total scale on ASSIS range from 36 (when the score for each item is 1) to 180 (when the score for each item is 5). Higher score on the scale predicts higher level of stress among students and lower score shows less severe acculturative stress. The score on the subscales can be calculated by totaling the individual score on the related items. "When the questionnaire at issue is reliable, people completely identical—at

**Table 1. Demographic information of the participants background of sample.**

| Gender | | Level | | Finance from | | Discipline | |
|---|---|---|---|---|---|---|---|
| Male | 152 (74.9%) | Undergraduate | 25 (12.3%) | CSC (China) | 173 (85.22%) | Natural Sciences | 50 (24.63%) |
| Female | 51 (25.1%) | Masters | 57 (28.1%) | HEC (Pakistan) | 14 (6.89%) | Social Sciences | 58 (28.57%) |
| | | PhD | 121 (59.6%) | Self | 16 (7.88%) | Engineering | 59 (29.06%) |
| | | | | | | Medical | 26 (12.80%) |

Note. CSC (Chinese Scholarship Council), HEC (Higher Education Commission of Pakistan).

**Table 2. Demographic information of interview participants.**

| S.NO | Code | Age | Gender | University | Level of education | Discipline | Fin. Status | Duration |
|------|------|-----|--------|------------|--------------------|-----------|-------------|----------|
| 1 | P-1 | 30 | M | HUST | PhD | Physics | HEC | 2nd Year |
| 2 | P-2 | 25 | M | HUST | Masters | Management | CSC | 2nd Year |
| 3 | P-3 | 27 | F | HUST | PhD | Chemistry | CSC | 2nd Year |
| 4 | P-4 | 21 | M | HUST | Undergraduate | Engineering | HEC | 2nd Year |
| 5 | P-5 | 25 | F | HUST | Masters | Economics | CSC | 1st Year |
| 6 | P-6 | 23 | M | CUG | Masters | Pol. Science | CSC | 1st Year |
| 7 | P-7 | 26 | M | CUG | PhD | Environ. Sc | CSC | 1st Year |
| 8 | P-8 | 25 | F | CUG | Masters | Geology | CSC | 1st Year |
| 9 | P-9 | 28 | M | CUG | PhD | Education | CSC | 2nd Year |
| 10 | P-10 | 31 | M | CUG | PhD | Management | HEC | 3rd Year |
| 11 | P-11 | 27 | M | HZAU | PhD | Biotechnology | CSC | 2nd Year |
| 12 | P-12 | 26 | M | HZAU | PhD | Botany | CSC | 2nd Year |
| 13 | P-13 | 23 | F | HZAU | Masters | Engineering | CSC | 1st Year |
| 14 | P-14 | 31 | M | HZAU | PhD | Economics | HEC | 3rd Year |
| 15 | P-15 | 24 | F | HZAU | Masters | Chemistry | CSC | 2nd Year |
| 16 | P-16 | 28 | M | WU | PhD | Finance | CSC | 3rd Year |
| 17 | P-17 | 30 | M | WU | PhD | Education | CSC | 3rd Year |
| 18 | P-18 | 32 | M | WU | PhD | Economics | CSC | 3rd Year |
| 19 | P-19 | 24 | F | WU | Masters | Management | CSC | 2nd Year |
| 20 | P-20 | 29 | M | WU | PhD | Economics | CSC | 2nd Year |

Note. P (participant), **HUST** (Huazhong University of Science and Technology, Wuhan), **CUG** (China University of Geosciences, Wuhan), **HZAU** (Huazhong Agriculture University, Wuhan), **WHU** (Wuhan University, Wuhan).

least with regard to their pleasure in writing–should get the same score, and people completely different score" [25]. Thus, questionnaire employed for this study is certainly reliable having an α score of 0.93. See Table 3.

For statistical analysis SPSS version 20 was used to identify the principal factors that cause acculturative stress. Principal Component Analysis (PCA) method was used. Employing Principal Component Analysis, the construct validity of the questionnaire can be verified [26]. If the questionnaire is validly constructed, its entire items signify underlying construct perfectly. Thus, total score on the 36 items on the Scale [22] represents one's interest and pleasure in writing accurately. PFA spots the construct–i.e. factors that trigger a dataset based on the correlation between variables [25, 27, 28]. The factor that elucidates the highest difference of variance, the variables segment is supposed to represent the underlying constructs but in PCA, it does not have the supposition of shared variance within the dataset [25, 27–29].

Conducting Principal Component Analysis, the sample size needs to be large enough [25, 27, 29]. According to [25] "the smaller size of sample has a bigger chance of correlation coefficients between items vary from the correlation coefficients between other samples". Generally, a researcher should have at least 10–15 participants per item. A good sample size requires 200–

**Table 3. Cronbach alpha score executed by SPSS 20.00.**

| Cronbach's Alpha | Cronbach's Alpha Based on Standardized Items | No. of Items |
|------------------|----------------------------------------------|--------------|
| .931 | .931 | 36 |

Note: Cronbach alpha score for reliability of ASSIS.

**Table 4. KMO test of sample adequacy based on correlation.**

| Kaiser-Meyer-Olkin Measure of Sampling Adequacy. | | .814 |
|---|---|---|
| Bartlett's Test of Sphericity | Approx. Chi-Square | 378.403 |
| | df | 15 |
| | Sig. | .000 |

Note: Kaiser-Meyer-Olkin Measure of Sampling Adequacy and Bartlett's Test of Sphericity for Sampling adequacy.

300, since this study has 203 sample-size so is considered fit for factor analysis (PCA). Kaiser-Meyer-Olkin measure of sampling adequacy (KMO) can perfectly indicate if the sample size is big enough to reliably extract factors [25]. KMO "signifies the ratio of the squared correlation between variables to the squared partial correlation between variables [25]. If the KMO is close to 1, a factor or factors can perhaps be extracted, since the conflicting pattern is visible. Therefore, KMO "values between 0.5 and 0.7 are mediocre, values between 0.7 to 0.8 are good, values between 0.8 to 0.9 are great and values above 0.9 are superb." [25]. For this study, the KMO values of the sample size is good at 0.81. See Table 4.

## Results

PCA on SPSS 20.00, extracted 7 factors of the ASSIS that account for total 68.53% of total explained variance. The total explain variance demonstrates the factors: Pakistani students expressed their concerns in the questionnaires. The factors with the high eigenvalues represent a factor among others with the high level of intensity. The more variance a factor holds, the more it causes acculturative stress. The factors with their eigenvalues, percentage of variance and cumulative percentages and extraction sums of squared loadings are given in Table 5.

The algebraic matrix calculations finally end up with eigenvalues [25]. It is displayed in Table 5; the eigenvalues represent a linear representation of the variance variables share. The longer an eigenvector is, the more variance it explains, the more importance it highlights [25, 30] It is measured on the rate of loadings on each variable on the eigenvector. Thus, Table 5 highlights factor 1: Perceived discrimination 68.53%, factor 2: homesickness 10.97%, and factor 3: Perceived Hate/ Rejection 9.15%. It can be concluded that Pakistani students showed concern on the mentioned factors.

In Table 6 all the seven 7 ASSIS Scales with their communalities have been presented. The results are matching and clearly recognizable. Commonalities of the factors show the values before and after extraction. PCA underlies the initial assumption that variance for all the factors are common; thus, before extraction the commonalities are all 1. So, the commonalities of

**Table 5. Summary of extracted factors of acculturative stress scale for Pakistani students.**

| Component | Extraction Sums of Squared Loadings | | | | |
|---|---|---|---|---|---|
| | Total | % of Variance | Total | % of Variance | Cumulative % |
| 1. Perceived Discrimination | 80.740 | 68.530 | 80.740 | 68.530 | 68.530 |
| 2. Homesickness | 12.925 | 10.970 | | | |
| 3. Perceived Hate/ Rejection | 10.785 | 9.154 | | | |
| 4. Fear | 6.193 | 5.257 | | | |
| 5. Culture Shock/Stress due to Change | 3.675 | 3.119 | | | |
| 6. Guilt | 3.046 | 2.585 | | | |
| 7. Miscellaneous/Nonspecific | .453 | .384 | | | |

Note: Summary of extraction and sums of loading on 7 scales of ASSIS.

**Table 6. Factors with commonalities.**

| Items | Raw | | Rescaled | |
|---|---|---|---|---|
| | Initial | Extraction | Initial | Extraction |
| 1. Perceived Discrimination | .815 | .304 | 1.000 | .373 |
| 2. Homesickness | 13.457 | 4.100 | 1.000 | .305 |
| 3. Perceived Hate/ Rejection | 18.493 | 9.115 | 1.000 | .493 |
| 4. Fear | 13.509 | 6.999 | 1.000 | .518 |
| 5. Culture Shock/Stress due to Change | 8.490 | 4.375 | 1.000 | .515 |
| 6. Guilt | 4.903 | 1.372 | 1.000 | .280 |
| 7. Miscellaneous/Nonspecific | 58.151 | 54.475 | 1.000 | .937 |

Note: ASSIS 7 factor with given commonalties after extraction.

the factors (Extraction column) reflect the common variance in the data structure and after extraction some factors are discorded. The amount of the percentage of variance on each factor can be evidently elucidated by the retained factors is presented after extraction column.

It is to be noted that according to [3] factor 7: Miscellaneous are not considered under any categories of the Scale, it has varied factors addressing the issues of international students. All the items of ASSIS Scale with their mean score, standard deviations and variance are presented orderly in Table 7.

In Table 7, mean score shows a significant concern of the Pakistani students on certain items. High mean score highlights the apprehension on the specific item area. For example, item 12: I miss my people and country of my origin, representing Homesickness, with 3.07 mean score, shows great concern among Pakistani students. Item 9: I feel sad leaving my relatives behind with a mean score of 3.62 is representing factor; Homesickness also validate the Pakistani student's concern. Thus, we can conclude that descriptive statistics on each item is in concordance with the PCA results. Table 6, represents the seven subscales as follows:

Perceived Discrimination Items (No.1-8), Homesickness Items (No.9-12), Perceived Hate/ Rejection Items (No.13-17), Fear Items (No.18-21), Culture Shock/ Stress due to Change Items (No. 22–24), Guilt Items (No 25–26), Miscellaneous/ Nonspecific Items (No. 27–36). These items are important because each of them separately focuses the core concerns of respondents.

## Discussion based on mixed method (concurrent triangulation strategy)

### On perceived discrimination

The results show that factor 1 Perceived Discrimination holds the highest percentage of variance (68.53%) which is the key concern of Pakistani students in China. A clarification for this perceived discrimination may comprise status loss [31], low self-esteem due to lack of family support [32], and culture shock [33], these are challenges generally international students face after relocation to a new cultural set up. It is also noted that underutilization of international students' knowledge and skill, [34], negative attitudes, and less compassion to variety of cultural values of the host culture by some Americans [35], concluded that international students are not warmly welcomed. The concern is in concordance with research studies suggesting that international students very often face the acculturative stress due the feeling of being rejected, alienated or discriminated against by members of the host culture [36]. Pakistani students concern indicates that they are passing through the challenges of alienation might lead

**Table 7. ASSIS scale items with mean, std. deviation and variance.**

| Item No. and Content. | Mean | Std. Deviation | Variance |
|---|---|---|---|
| I feel many opportunities are denied to me. | 3.0246 | 1.25248 | 1.569 |
| I am treated differently in social gatherings. | 3.0788 | 1.35840 | 1.845 |
| My colleagues are biased towards me. | 2.9113 | 1.25943 | 1.586 |
| I feel that I receive unequal treatment. | 2.5616 | 1.36066 | 1.851 |
| I feel that my community is discriminated here. | 2.5320 | 1.32108 | 1.745 |
| I am treated differently because of my race. | 2.4828 | 1.29487 | 1.677 |
| I am treated differently because of my color. | 3.1182 | 1.19645 | 1.431 |
| I am denied what I deserve. | 3.0936 | 1.36291 | 1.858 |
| I feel sad leaving my relatives behind. | 3.6207 | 1.18529 | 1.405 |
| Home sickness bothers me. | 3.3103 | 1.23370 | 1.522 |
| I feel sad living in unfamiliar surroundings. | 2.8621 | 1.24308 | 1.545 |
| I miss the people and country of my origin. | 3.7537 | 1.08929 | 1.187 |
| People show hatred towards me nonverbally. | 2.3547 | 1.17832 | 1.388 |
| People show hatred towards me verbally. | 2.0936 | 1.00794 | 1.016 |
| People show hatred to me through actions. | 2.1675 | 1.07239 | 1.150 |
| Others are sarcastic towards my cultural values. | 2.5813 | 1.20109 | 1.443 |
| Others don't like my cultural values. | 2.6552 | 1.20614 | 1.455 |
| I fear for my personal safety because of my different culture. | 2.6502 | 1.33158 | 1.773 |
| I generally keep low profile due to fear. | 2.5665 | 1.15152 | 1.326 |
| I feel insecure here. | 2.0394 | 1.18924 | 1.414 |
| I relocate frequently for fear of others. | 2.2562 | 1.11843 | 1.251 |
| Multiple pressures are placed on me after migration. | 2.7783 | 1.18369 | 1.401 |
| I feel uncomfortable to adjust to new food. | 3.5517 | 1.21502 | 1.476 |
| I feel uncomfortable to adjust to new cultural values. | 3.0000 | 1.22676 | 1.505 |
| I feel guilty to leave my family and friends behind. | 2.5468 | 1.24341 | 1.546 |
| I feel guilty that I am living a different life style here. | 2.3645 | 1.17529 | 1.381 |
| I feel nervous to communicate in Chinese language. | 3.1478 | 1.23794 | 1.533 |
| I feel hesitant to participate in social gatherings with Chinese. | 2.6847 | 1.21424 | 1.474 |
| I feel angry that my people are considered inferior here. | 3.5271 | 1.13583 | 1.290 |
| It hurts me that people don't understand my cultural values. | 3.0493 | 1.20541 | 1.453 |
| I feel low because of my different cultural background. | 2.6059 | 1.19502 | 1.428 |
| I don't feel a sense of belonging here. | 2.8867 | 1.13973 | 1.299 |
| I feel lonely. | 3.0049 | 1.30307 | 1.698 |
| I realize that people don't associate with me because of my ethnicity. | 2.6946 | 1.23293 | 1.520 |
| My interaction with Chinese is unsatisfactory because of language incompetency. | 3.4335 | 1.28173 | 1.643 |
| I worry about my future whether to stay here or go back my home country. | 3.2414 | 1.28428 | 1.649 |

Note: Descriptive values of ASSIS items including mean score, standard deviations and variance.

them to their choice of engagement and the style of socialization with people who are not members of the host culture for social support [37]. Close social interaction with the host culture students helps international students harness the challenges of acculturation process [41] sojourners' length of residence greatly affects their adaptation. The degree that sojourners' contact with the co-nationals can create an impact on their adaptation to the host culture. As P-15 narrated, "when I first came I was so lonely but in the very first month ISO (international students office) arranged orientation classes, city tour and a Chinese language class, which made me confident and I made some good friends too, I appreciate my university's role that

help new comers to be confident, familiar with host culture and local students." This effective support by Chinese universities play a crucial role in adopting to new cultural setting, academic environment and lowering acculturative stress among international students.

The factor "Perceived Discrimination" as the most concerned factors based on quantitative results loaded with 68% variance and stood at first place among all the seven factors of the ASSIS scale. The items of the scale under discrimination as its factor, are all close ended questions and the participants cannot fully express their perception. They can only answer as YES or NO but cannot explain in depth the severity and nature of their feelings.

Qualitative data findings provided this factor with deep insight as the interviewee discussed in detail on discrimination. Mix Method Research (MMR) provided the data with effective triangulation to confirm and disconfirm the finding gotten from both the sources of data collection. Qualitative data disconfirmed surprisingly the most leaded factor "Perceived Discrimination" which was stood at top most concern of Pakistani students in China. After mixing qualitative and quantitative findings, it can be concluded that this discrimination is not the discrimination as negative but it's a natural perception of many international students because of their unfamiliarity with the host country laws and cultural norms. The more the students get familiar with the new environments either social or academic, the more students feel secured and equally treated [41]. The quantitative feelings of discrimination were because of students' expectations based on the previous social and academic backgrounds.

## On homesickness

Homesickness subscale falls at the second most concerned factor among the students. Homesickness subscale with a (10.97%) percentage of total variance indicates that most of Pakistani students are suffering from homesickness. Several studies highlight that international students express their feelings of loneliness due to leaving relatives, friends and family members in the countries of their origin as they migrated to a foreign country to pursue their higher education [38]; Less sociability and sense of acquaintance of local students with Pakistani students also stimulating the feelings of homesickness. The Chinese with good English communication and understanding feel confident and make friendship with international students and Pakistani students with Chinese language skills often feel free to interact with their Chinese class mates. [40] found that students in the initial stage of mobility, face difficulty in social interaction due to unproficiency in Chinese language which further lead these students to isolation and stressful life. Factor 7 of the also addresses different concerns of the international students (see Table 6) such as Item 27; My interaction with Chinese is unsatisfactory because of my language incompetency (mean score 3.43), demonstrate Pakistani students concerns over language barriers causing stress. Qualitative findings record a significant concern of Pakistani students on homesickness. The findings show homesickness severity in the students who are in the first year of their studies. Interview participants who study in their 2nd and 3rd year are expressing a very mild concern unlike 1st year students. As P-13 a 1st year student narrated, "This is my first journey out of my home and sometimes I badly miss my family. I am far off my family members and relatives but my objectives motivate me to face it patiently". Foreign students' feelings of obligation to keep attach to the roots of their own culture [39] stimulate homesickness. The response of a 3rd year study participant P-18 was as, "I remember my initial days, I was feeling lonely missing my family and people of my culture. Gradually, I began to interact with locals, other international students and my country-mates so if you ask me today, I am the part of this society, I feel happy and I am focusing on my studies". [41] revealed that the low level of willingness to interact with host people results in lack of knowledge about the host country and lower sociocultural adaptation. Home sickness is a common phenomenon among

all the international students who migrate voluntarily abroad. Pakistani students like others are motivated towards their mobility to China and abroad experiences so are expected to overcome the feeling of home sickness by resilience and determination.

## On perceived hate

Perceived Hate holds (9.15%) of variance as a concern of Pakistani students which was recorded moderate and in uniform scores on each item of the factors such as (see Table 6) Item 13 (mean 2.35) "People show hatred toward me non-verbally", Item 14 (mean 2.09) "People show hatred toward me verbally", Item 15 (mean 2.16) "People show hatred toward me through action", Item 16 (mean 2.58) "Others are sarcastic towards my cultural values", and Item 17 (mean 2.65) "Others don't like my cultural values". These items record a mild concern of Pakistani students because of their insufficient social interaction during initial days of migration. Similar response was noted during interview of P13, "University facilitates students to accommodate their guests in universities' exchange centers but this was all booked, and when I went to book a room for my guest outside I was denied, saying we don't entertain foreigners, that was a bitter experience and I perceived that I am being hated as a foreigner." It is to be noted that Chinese government issue licensed to hotels and categorize them on the basis of their facilities so that a clean and peaceful environment should be ensured, the hotels with the facilities provided can host foreigners. These findings are in concordance with the previous study conducted by [41] that the students very positively evaluate their intercultural experiences, especially with regards to their relationships with Chinese people. The perception of being hated in China was due to newly migrated students' unawareness of culture and local rules and regulations. The feelings became lower with the duration of stay in China as mentioned by [40] in study on intercultural adaptation of Pakistani students in China.

## On fear

Fear was found among Pakistani students but this feeling of fear seems to be due to different culture, different ways of life, different language and unfamiliar surroundings. Individual personality type and first-time abroad experience also intensify feeling of fear in Pakistani students. The feelings of fear is not on the scale of severe intensity, it might be because Pakistan and China have brotherly socio-political relations that encourages international students from Pakistan and this feelings of trusted friendship lowers the sense of insecurity like many confirms in US for increasing number of crimes, racial discrimination and socio-political realities of off and on hostile relations among some international students' country of origins (Iran, Iraq, etc.) and the United States [3]. As stated earlier, fear is a concern of international students in some of the western societies where students are discriminated on the basis of religion, color and political issues. Interview feedbacks show that these students are feeling secure in China as when inquired whether has she ever sensed a threat or fear during your stay here, P-15 responded, "Being a female from a conservative family background, I and my mother were both feeling insecure to go abroad, but I feel equal, powerful and secure". The qualitative data concludes that Pakistani students in Chinese universities are recording no concern on fear or insecurity. The results show that feelings in insecurity in a new socio-cultural and academic environment lead to the feelings of fear among students. The results of this study support the findings of [41] which stated that possessing Chinese language proficiency provided students with confidence in different social and cultural [40] as well as in academic settings that ultimately resulted in lowering the acculturative stress, anxiety and depression.

## On culture shock/stress due to change

Culture shock subscale recorded concerns of Pakistani students. All the 3 items show that culture shock is generated due to the pressure placed on the newly arrived international students where everything is unique, different for what they had experienced in their home countries. The findings confirm that Pakistani students have a critical concern and are experiencing culture shock. This culture shock or stress due to change leads these students to acculturative stress. Qualitative data demonstrate three major concerns on factors like physiological, interpersonal and psychological that further contribute to mental strain and culture shock. The symptoms commonly related with culture shock were buoyed to varying grades for each of the participants by the data collected from the interviews. The participants, in spite of expressed or implied feelings of confusion or astonishment from prior in their 1st year which is in concordance with [40], were overall considerate about any cultural variances rather than contemptuous, to the degree that by the 3rd year, they stated that any culture shock they had experienced, if at all, was marginal and in the past. P-5, for example, stated: "I noticed very few differences, but they are easy to be accepted and are not stressful." P-13 declared that "I didn't notice any sign of culture shock which I could remember. I was readily accepting new ideas and were struggling to adapt." Finally, P-18, when asked if there is anything that shocked him in current year, responded, "Maybe in my initial days in China, which I don't remember." Culture Shock is the most researched topic in the literature on foreign students [31].

## On guilt

The factor "Guilt" captured (2.28%) of total variance, shows a slight concern of guilt among Pakistani students studying in China. (See Table 6) Item 25, (mean 2.54) "I feel guilty leaving my friends and family behind", Item 26 (mean 2.36) "I feel guilty that I am living a different life style here". Adjusting to the host culture is generally conceived infidelity and betrayal to their own culture. This is a natural phenomenon that one loves one's own culture, family and homeland. This has been revealed by respondent upon asking whether he has felt guilt, P-10 stated, "I, during my first year, was many times totally overwhelmed by severe sense of guilt... To leave my kids and family behind." International students seemingly struggle to maintain their identity in the host culture but the dominant host culture somehow influence them to adapt. As P-18 asserted, "I like my own ways of life, China is totally different, to adapt certain ways like eating habits and time at first made me stressful but I tried to forget my past and finally adjusted". The dominance of the host culture and the pressure of acculturation that is crucial for their academic excellence, self-esteem and physical well-being tempt them to adapt the host culture but morally it seems to them as insincere to their own culture and people.

## Miscellaneous such as language inability and food adaptation

Some very important concerns of Pakistani students in China were highlighted by Item 27, "I feel nervous to communicate in Chinese" (mean 3.14) indicates the intense concern of Pakistani students on the language incompetency. This highly loaded concern with a significant mean score indicates the language is undoubtedly a barrier in acculturation process either social or academic. It deters the process of adaptation and gradually leads to many disorders like acculturation stress, depression and sometimes inferiority complex. Item 29, "I feel angry that my people are considered inferior here (mean 3.52). The mean score for this item shows that students develop feelings of loneliness because they feel themselves homesick and in minority. Findings from interviews demonstrate four significant concerns of Pakistani students. R-7 on being asked about its effects stated, "I don't exactly know. . .but when I don't get enough sleep at night, I feel sleepiness during the day, general tiredness, irritability and

problems with concentration." For language barriers the concerns of the 1st year students were higher as compared to 2nd and 3rd year students. P-7 a 1st year PhD student narrated, "I understand Chinese is very important. . . but it's my first year and this language is very difficult to learn." Chinese language incompetency cause fear, anxiety and is a key factor hindering academic, psychological and socio-cultural adaptation. It negatively affects Pakistani students because [41] language barriers cause difficulty in relationship with professor, staff and peers and understating and adjusting to the process of learning in Chinese educational settings. Chinese language has characters construction unlike alphabetic so that is something new and seems hard to be acquired. When asked P-16 if he had experienced any food issue, responded, "Last year I went to another province to attend a conference, the food was so different and too spicy for me. I tried but couldn't." 1st year students' responses on interview script show their concern on loneliness. As P-5, a 1st year PhD student stated, "I was feeling lonely because I had no any friend local or international but after 3 months, I found my country-mates, foreign students and even get acquainted with my Chinese class mates there is no more loneliness." This statement clearly demonstrates that social interaction on the part of international students is crucial to overcome acculturative stress.

## Conclusion

In quantitative segment, the study administered ASSIS Scale to examine acculturative stress among Pakistani Students studying in China. The quantitative findings of each subscale of the ASSIS and total variance on some factors indicate that Pakistani students are going through a mild acculturative stress and it can be generalized that the more the variance percentage grows up, the more we perceive the level of acculturative stress. ASSIS includes 7 subscales or factors that are assumed to be responsible for acculturative stress among international students. Through these factors, we understand that Perceived Discrimination, Perceived Hate, Homesickness, Fear, Culture Shock/ Stress due to Change and Guilt are the significant factors causing acculturation stress among Pakistani students studying in China. Apart from aforementioned factors, food, language, lacking sense of belonging and loneliness are important factors to be considered responsible for causing acculturative stress.

The qualitative findings illustrate that Pakistani students in China are expressing their major concerns on culture shock, homesickness, food and language barriers while disconfirm ASSIS findings like perceived discrimination, hate, fear and guilt as factors responsible for acculturative stress. The study findings indicate that many factors are affecting international students because of language incompetency. Therefore, we suggest that the sending country should provide an effective literature of host country's culture and engage the aspirants to learn Chinese before leaving their country. The study suggested that pre-migration orientation lectures about host country's cultural values and campus environment, and post-migration extra-curricular, cultural activities and maximum social interaction with local students can effectively acculturate students in new cultural setting, and can lower their acculturative stress.

Findings of the present study could be used for, and guide the support mechanism at universities for international students in Chinese higher education intuitions.

Firstly, in relation to the orientation and counseling, the materials provided to the students at the pre-departure stage should comprise of basic literature that should be clear and up to the date to allow aspirants understand the requirements for their future study. It is will be more appropriate and productive if university should integrate and invite senior international students for lectures to share their experience and guide the newly arrived international students. A comprehensive orientation by elder students and university staff would be more effective to cultivate readiness among new students to cope with the unfamiliar social and academic

setting. It is more practical way of orientation because the seniors can share their lived experiences and advise them how to respond to these difficulties and stressors. Furthermore, orientation courses should include Chinese language exercises as well as there should be arranged classes of English language to improve the academic writing skills of Pakistani students as English is not their first language.

Secondly, academic adaptation and understanding laws of Chinese higher education are a key apprehension of most of the participants of the study. It has also been shown in the present study, in many of the occurred cases, the unawareness of rules and degree requirements negatively impact the students' adaptation both academically as well as socially. Most of them are seeking information and guidance through peers including senior countrymates and other international students, which means that they do not receive adequate and timely information of their requirements. The university ought to provide more information regarding rules and regulations of the university as well as rule amended by the ministry at any time of the academic year so that these students should be mentally prepare to respond to the requirements positively. The ISO and the secretary of each school should work on the effectiveness of dissemination of information to every student. WeChat groups might be generated for the students, because social media approach seems to have more positive effect on the students' academic and social adaptation.

Thirdly, it is crucial for the university to produce more chances and form more co-curricular activities to interact and integrate Pakistani students, other international students and Chinese students. Effective strategies should be applied to stimulate maximum international student to participate efficiently in cultural activities.

Fourthly, the staff and faculty should recognize the key weakness and differences in the students' previous educational background as well as academic language abilities. Therefore, faculty should have a proper understanding student's needs so as to offer effective supervision and instruction. It would be more effective to plan the courses by opening with outline of the particular discipline, which familiarizes each of the students with, educational norms and important ideas of important application to the major of study and academic standards.

Finally, the supervisors should settle their availability in advance as schedule and should encourage the students to seek guidance. Similarly, it is significantly advised for international students to seek the assistance of supervisors and staff while needed and be positive while facing academic challenges. Generally, Pakistani students might straightforwardly acclimate to Chinese academic and social norms and practices and may flourish in their instructions provided that university leadership, faculty, and students themselves meticulously comprehend conceivable stressors as well as generate and withstand advantageous academic environments.

## Limitation of the study

The findings are significant but covering only 5 provinces with the sample 203 can be further studied covering more provinces and large sample based on different demographic characteristics like marital status, country of origin, religion and personality type is recommended. To generalize the findings to all international students' needs care because home culture of Pakistan may not be similar to many other countries.

## Supporting information

**S1 File.**
(SAV)

## Author Contributions

**Conceptualization:** Mudassir Hussain.

**Data curation:** Mudassir Hussain.

**Methodology:** Mudassir Hussain.

**Software:** Ghulam Raza Sargani.

**Writing – original draft:** Mudassir Hussain.

**Writing – review & editing:** Cao Shan.

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
