## [Decision Letter · Decision Letter 0]

20 Jul 2020

PONE-D-20-18965

A Mix-Method investigation on Acculturative Stress among Pakistani Students in China

PLOS ONE

Dear Dr. Hussain,

Thank you for submitting your manuscript to PLOS ONE. After careful consideration, we feel that it has merit but does not fully meet PLOS ONE’s publication criteria as it currently stands. Therefore, we invite you to submit a revised version of the manuscript that addresses the points raised during the review process.

We look forward to receiving your revised manuscript.

Kind regards,

Sakamuri V. Reddy

Academic Editor

PLOS ONE

Journal Requirements:

4. Your ethics statement must appear in the Methods section of your manuscript. If your ethics statement is written in any section besides the Methods, please move it to the Methods section and delete it from any other section. Please also ensure that your ethics statement is included in your manuscript, as the ethics section of your online submission will not be published alongside your manuscript.

Reviewers' comments:

Reviewer's Responses to Questions

**Comments to the Author**

1. Is the manuscript technically sound, and do the data support the conclusions?

Reviewer #1: No

Reviewer #2: Yes

2. Has the statistical analysis been performed appropriately and rigorously? 

Reviewer #1: No

Reviewer #2: Yes

3. Have the authors made all data underlying the findings in their manuscript fully available?

Reviewer #1: No

Reviewer #2: Yes

4. Is the manuscript presented in an intelligible fashion and written in standard English?

Reviewer #1: No

Reviewer #2: Yes

5. Review Comments to the Author

Reviewer #1: Major points:

1. Importance. The authors might explain in detail the importance and implications of this study. Specifically, why does this paper focus on Pakistani students in China? Does this paper have generalizable implications on international students’ acculturative stress of students from other countries and/or studying in countries other than China?

2. Survey. More details about the survey are needed: when was the survey implemented, and how? What is the sampling approach? Is the sample representative (or biased)? How was the sample size determined?

3.Quantitative analysis. It is unclear about the method as there is no such a method called Principal Factor Analysis – I guessed that you meant Principal Component Analysis (PCA). It is also unclear why factor analysis is used to assess the relative importance of different stress measures, which I think is inappropriate. I would correlate the generated factor measure with other student-level and institution-level covariates to examine the heterogeneity in the acculturative stress.

4. Qualitative analysis. This paper has done a nice job using a mixed method study. However, there lacks compelling explanations on why the findings on perceived discrimination et al. differ between the quantitative and qualitative analyses.

5. The policy recommendations are a little bit disconnected from the evidence presented in this paper. There needs to be more evidence on the potential effects of the orientation lectures – just one quote from the interview is not enough.

Minor points:

1. This paper is unfortunately poorly written. The authors might want to carefully edit the manuscript. There are many cases of grammar mistakes, repeated commas, and spaces between words. For example, “investigation” and “among” in the title should be capitalized.

2. Tables and Figures should be formatted. Labels should be added.

3. Some translation problems. For example,” the standard translation of Chinese global development strategy you mentioned should be “One Belt, One Road” rather than “Chinese Belt and Road.”

Reviewer #2: 1. Literature review:Add some literature pertaining to Pakistani international students. The following literature can be seen

1. Noreen, Sehrish, Fan Wei Wei, Mehvish Zareen, and Sameena Malik. 2019. "The Intercultural Adjustment of Pakistani students at Chinese Universities." INTERNATIONAL JOURNAL OF ACADEMIC RESEARCH IN BUSINESS AND SOCIAL SCIENCES 9 (3).

2. Su, Xiaoqing. 2017. "The Intercultural Adaptation of the Pakistani Students at Chinese Universities." Universal Journal of Educational Research 5 (12): 2236-2240.

2. In Research Questions, add a question pertains to role of language in cultural adjustment.

3. The study was conducted with purpose to explore the role of language, academic system and culture in the adjustment of Pakistani students in Chinese students. Keeping in view, add some content related to these areas n results /conclusion section.

4. In the results section, the current study can use the available literature on Pakistani students' adjustment in Chinese (referred above) to compare /support the findings of the current study.

6. PLOS authors have the option to publish the peer review history of their article (what does this mean?). If published, this will include your full peer review and any attached files.

Reviewer #1: No

Reviewer #2: **Yes: **Asma Bashir

---

## [Author Response · Author response to Decision Letter 0]

13 Aug 2020

Reviewer #1: Major points:

1. Importance. The authors might explain in detail the importance and implications of this study. Specifically, why does this paper focus on Pakistani students in China? Does this paper have generalizable implications on international students’ acculturative stress of students from other countries and/or studying in countries other than China?

Response 1 

Conferring to the statistics announced by Chinese Ministry of Education in 2020 and According to Asia Pacific Daily Report there are more than 30,000 Pakistani students who are studying in various Chinese universities, in 2020 which makes Pakistan the 3rd largest source of international students in the China. As the number of Pakistani students on Chinese campuses has risen, Currently, 6,156 Pakistani students are studying in Ph.D., 3,600 in Masters, 11,100 in Bachelors and 3,000 in Short Term Exchange Programs across China for their researches and immigrants adapt to a new cultural environment and on ways Chinese hosts can help the sojourners adapt to their new environment have become of great significance. [Su, Xiaoqing. 2017. "The Intercultural Adaptation of the Pakistani Students at Chinese Universities." Universal Journal of Educational Research 5 (12): 2236-2240].

2. Survey. More details about the survey are needed: when was the survey implemented, and how? What is the sampling approach? Is the sample representative (or biased)? How was the sample size determined?

Response 2

In the present study the simple random sampling was applied whereas in a simple random sample, every member of the population has an equal chance of being selected, which includes the whole population. 

To conduct this type of sampling, it can use tools like random number generators or other techniques that are based. Hence the sampling is the statistical process of selecting a subset (called a “sample”) of a population of interest for purposes of making observations and statistical inferences about that population. While Social science research is generally about inferring patterns of behaviors within specific populations. Therefore, we cannot study entire populations because of feasibility and cost constraints, and henceforth, we have selected a representative sample from the population of interest for observation and analysis. It is extremely important to choose a sample that is truly representative of the population so that the inferences derived from the sample can be generalized back to the population of interest. The improper and biased sampling is the primary reason for often divergent and erroneous inferences reported in opinion polls and exit polls conducted by different polling groups entirely on chance.

3.Quantitative analysis. It is unclear about the method as there is no such a method called Principal Factor Analysis – I guessed that you meant Principal Component Analysis (PCA). It is also unclear why factor analysis is used to assess the relative importance of different stress measures, which I think is inappropriate. I would correlate the generated factor measure with other student-level and institution-level covariates to examine the heterogeneity in the acculturative stress.

Response 3

The abbreviation used as PFA was a clerical mistake and is replaced with PCA- Principal Component Analysis...

Factor analysis is a technique used to reduce a large number of variables to a smaller number of factors. This technique extracts the largest common variance from all variables and puts them into a common score. As an index of all variables, we can use this score for further analysis. Factor analysis is part of the General Linear Model (GLM). The method also assumes the following assumptions: there is a linear relationship, there is no multicollinearity, related variables are included in the analysis, and there is a true correlation between the variables and the factors. There are several methods available, but the most commonly used is principal component analysis. Principal component analysis: This is the most common method used by researchers. PCA starts extracting the maximum variance and puts them into the first factor. After that, it removes that variance explained by the first factors and then starts extracting maximum variance for the second factor. This process goes to the last factor.

4. Qualitative analysis. This paper has done a nice job using a mixed method study. However, there lacks compelling explanations on why the findings on perceived discrimination et al. differ between the quantitative and qualitative analyses.

Response 4 

Quantitative variables were expressed as means and its related standard deviations while qualitative variables were presented in the form of frequency and percentages. Logistic regression analysis was done for severely stressed students versus others as a dependent variable with each of age, gender, Bachelor of Allied Health Sciences versus others, and living with parents versus others. Factor analysis techniques with varimax rotation with Kaiser Normalization criteria was used to discover hidden factors for stress. Level of significance was set at 0.05 throughout the study.

The findings after convergence of both quantitative and qualitative data, differ based on results which led authors to surprising facts, details have been added to factor (Perceived Discrimination)

5. The policy recommendations are a little bit disconnected from the evidence presented in this paper. There needs to be more evidence on the potential effects of the orientation lectures – just one quote from the interview is not enough.

Response 5 

 A detail on recommendations based on the study findings has been added which can clarify policy recommendations, the evidences are taken from converged data analysis and findings.

Minor points:

1. This paper is unfortunately poorly written. The authors might want to carefully edit the manuscript. There are many cases of grammar mistakes, repeated commas, and spaces between words. For example, “investigation” and “among” in the title should be capitalized.

Response 1

The paper has been read carefully and made all the grammatical corrections such as commas and capitalizations etc.

2. Tables and Figures should be formatted. Labels should be added.

Response 2

Tables and figures have been labeled.

3. Some translation problems. For example,” the standard translation of Chinese global development strategy you mentioned should be “One Belt, One Road” rather than “Chinese Belt and Road.”

Response 3

Translations of terms from Chinese to English has been paid much attention, Chinese Belt and Road as mentioned has been replaced with One Belt One Road.

Reviewer #2: 1. Literature review:

1. Add some literature pertaining to Pakistani international students. The following literature can be seen

Response 1

The authors have tried their best to find related literature on Pakistani students academic, socio-cultural and psychological adaptation but there found very less pertaining to this particular group of international students in China as mentioned in literature review parts. The already cited 2 articles have been referenced already. Therefore, authors included literature done on general international students in China and-abroad. 

1. Noreen, Sehrish, Fan Wei Wei, Mehvish Zareen, and Sameena Malik. 2019. "The Intercultural Adjustment of Pakistani students at Chinese Universities." INTERNATIONAL JOURNAL OF ACADEMIC RESEARCH IN BUSINESS AND SOCIAL SCIENCES 9 (3).

2. Su, Xiaoqing. 2017. "The Intercultural Adaptation of the Pakistani Students at Chinese Universities." Universal Journal of Educational Research 5 (12): 2236-2240.

2. In Research Questions, add a question pertains to role of language in cultural adjustment.

Response 2

A question (What is the role of language in causing acculturative stress?) has been added to Research Questions Part as suggested by Reviewer 2.

3. The study was conducted with purpose to explore the role of language, academic system and culture in the adjustment of Pakistani students in Chinese students. Keeping in view, add some content related to these areas n results /conclusion section.

Response3.

The conclusion section has been modified by adding content that is based on study results and-findings.

4. In the results section, the current study can use the available literature on Pakistani students' adjustment in Chinese (referred above) to compare /support the findings of the current study.

Response 4

The authors tried to comprehensively portrayed the available literature on Pakistani students in China as well as literature available on other international students in China and abroad to provide a general understating on acculturation experiences of international students as general and Pakistani students in particular...

Thank You for your valuable comments, the review was extremely useful and helped me refined my article and enhanced my learning.

Sincerely Yours

Dr Mudassir Hussain

---

## [Decision Letter · Decision Letter 1]

27 Aug 2020

PONE-D-20-18965R1

A Mix-Method Investigation on Acculturative Stress Among Pakistani Students in China

PLOS ONE

Dear Dr. Hussain,

Thank you for submitting your manuscript to PLOS ONE. After careful consideration, we feel that it has merit but does not fully meet PLOS ONE’s publication criteria as it currently stands. Therefore, we invite you to submit a revised version of the manuscript that addresses the points raised during the review process.

ACADEMIC EDITOR: As the reviewer-2 noted below, the authors are advised to address previous studies conducted on Pakistani students in China in revising the manuscript.

We look forward to receiving your revised manuscript.

Kind regards,

Dr. Sakamuri V. Reddy

Academic Editor

PLOS ONE

Reviewers' comments:

Reviewer's Responses to Questions

**Comments to the Author**

1. If the authors have adequately addressed your comments raised in a previous round of review and you feel that this manuscript is now acceptable for publication, you may indicate that here to bypass the “Comments to the Author” section, enter your conflict of interest statement in the “Confidential to Editor” section, and submit your "Accept" recommendation.

Reviewer #2: (No Response)

2. Is the manuscript technically sound, and do the data support the conclusions?

Reviewer #2: Yes

3. Has the statistical analysis been performed appropriately and rigorously? 

Reviewer #2: Yes

4. Have the authors made all data underlying the findings in their manuscript fully available?

Reviewer #2: Yes

5. Is the manuscript presented in an intelligible fashion and written in standard English?

Reviewer #2: Yes

6. Review Comments to the Author

Reviewer #2: A part of literature review must address previous studying conducted on Pakistani students in China, and what these studies have explored. The same can be used to argue/support the findings in the discussion. However, all this is missing in the current article.

7. PLOS authors have the option to publish the peer review history of their article (what does this mean?). If published, this will include your full peer review and any attached files.

Reviewer #2: **Yes: **Asma Bashir

---

## [Author Response · Author response to Decision Letter 1]

17 Sep 2020

1. A part of literature review must address previous studying conducted on Pakistani students in China, and what these studies have explored. The same can be used to argue/support the findings in the discussion. However, all this is missing in the current article.

Response 1 

Literature review section of the article has been modified and revised based on the previous literature found and suggested by respected reviewer 2. The below mentioned two empirical studies were referred to and cited in the literature review part as well as discussed, their findings in the discussion part of this study where needed. The two previous studies had found the factor like language barriers, new academic and social environment, cultural differences responsible for hindering intercultural adaptation but ultimately students learned effective coping strategies and student’s length of stay in China played a crucial role in overcoming the issues of anxiety and depression in China. Similarly, as this Mix Method Study mainly focuses on Pakistani Students acculturations and aims to see the level of stress due to acculturation among Pakistani students in China, mostly comes up with the similar findings and in concordance with the findings like language acquisition provides new students with confidence that enhance social and academic life of students in a cross-cultural environment. 

1. Noreen, Sehrish, Fan Wei Wei, Mehvish Zareen, and Sameena Malik. 2019. "The Intercultural Adjustment of Pakistani students at Chinese Universities." INTERNATIONAL JOURNAL OF ACADEMIC RESEARCH IN BUSINESS AND SOCIAL SCIENCES 9 (3).

2. Su, Xiaoqing. 2017. "The Intercultural Adaptation of the Pakistani Students at Chinese Universities." Universal Journal of Educational Research 5 (12): 2236-2240.

---

## [Editor Report · Decision Letter 2]

21 Sep 2020

A Mix-Method Investigation on Acculturative Stress Among Pakistani Students in China

PONE-D-20-18965R2

Dear Dr. Hussain,

We’re pleased to inform you that your manuscript has been judged scientifically suitable for publication and will be formally accepted for publication once it meets all outstanding technical requirements.

Kind regards,

Dr. Sakamuri V. Reddy

Academic Editor

PLOS ONE

---

## [Editor Report · Acceptance letter]

24 Sep 2020

PONE-D-20-18965R2 

A Mix-Method Investigation on Acculturative Stress Among Pakistani Students in China 

Dear Dr. Hussain:

I'm pleased to inform you that your manuscript has been deemed suitable for publication in PLOS ONE. Congratulations! Your manuscript is now with our production department. 

Kind regards, 

on behalf of

Dr. Sakamuri V. Reddy 

Academic Editor

PLOS ONE